# Improvement of Selected Morphological, Physiological, and Biochemical Parameters of Roselle (*Hibiscus sabdariffa* L.) Grown under Different Salinity Levels Using Potassium Silicate and *Aloe saponaria* Extract

**DOI:** 10.3390/plants11040497

**Published:** 2022-02-11

**Authors:** Alaa Idris Badawy Abou-Sreea, Mohamed H. H. Roby, Hayam A. A. Mahdy, Nasr M. Abdou, Amira M. El-Tahan, Mohamed T. El-Saadony, Khaled A. El-Tarabily, Fathy M. A. El-Saadony

**Affiliations:** 1Horticulture Department, Faculty of Agriculture, Fayoum University, Fayoum 63514, Egypt; aib00@fayoum.edu.eg; 2Department of Food Science and Technology, Faculty of Agriculture, Fayoum University, Fayoum 63514, Egypt; mhr00@fayoum.edu.eg; 3Botany Department, National Research Centre, Dokki, Giza 12622, Egypt; hayam1111@yahoo.com; 4Soil and Water Department, Faculty of Agriculture, Fayoum University, Fayoum 63514, Egypt; nma02@fayoum.edu.eg; 5Plant Production Department, Arid Lands Cultivation Research Institute, The City of Scientific Research and Technological Applications, SRTA-City, Borg El Arab, Alexandria 21500, Egypt; amira_eltahan2012@yahoo.com; 6Department of Agricultural Microbiology, Faculty of Agriculture, Zagazig University, Zagazig 44511, Egypt; m.talaatelsadony@gmail.com; 7Department of Biology, College of Science, United Arab Emirates University, Al-Ain 15551, United Arab Emirates; 8Agricultural Botany Department, Faculty of Agriculture, Zagazig University, Zagazig 44511, Egypt

**Keywords:** anthocyanin, *Hibiscus sabdariffa* L., potassium, salinity, silicon

## Abstract

Two successive field trials were carried out at the experimental farm of the Agriculture Department of Fayoum University, Fayoum, Egypt, to investigate the sole or dual interaction effect of applying a foliar spray of *Aloe saponaria* extract (Ae) or potassium silicate (KSi) on reducing the stressful salinity impacts on the development, yield, and features of roselle (*Hibiscus sabdariffa* L.) plants. Both Ae or KSi were used at three rates: 0% (0 cm^3^ L^−1^), 0.5% (5 cm^3^ L^−1^), and 1% (10 cm^3^ L^−1^) and 0, 30, and 60 g L^−1^, respectively. Three rates of salinity, measured by the electrical conductivity of a saturated soil extract (ECe), were also used: normal soil (ECe < 4 dS/m) (S1); moderately-saline soil (ECe: 4–8 dS/m) (S2); and highly-saline soil (ECe: 8–16 dS/m) (S3). The lowest level of salinity yielded the highest levels of all traits except for pH, chloride, and sodium. Ae at 0.5% increased the values of total soluble sugars, total free amino acids, potassium, anthocyanin, a single-photon avalanche diode, stem diameter, fruit number, and fresh weight, whereas 1% of Ae resulted in the highest plant height, chlorophyll fluorescence (Fv/Fm), performance index, relative water content, membrane stability index, proline, total soluble sugars, and acidity. KSi either at 30 or 60 g L^−1^ greatly increased these abovementioned attributes. Fruit number and fruit fresh weight per plant also increased significantly with the combination of Ae at 1% and KSi at 30 g L^−1^ under normal soil conditions.

## 1. Introduction

Roselle (*Hibiscus sabdariffa* L.) is a medicinal herb from the Malvaceae family. It is an erect, mostly-branched, annual summer shrub with a deep penetrating taproot and variable-colored, green-to-red leaves and large, short-peduncled, red-to-yellow flowers with a dark center [1]. Roselle is productively cultivated in the tropics and subtropics [2].

The young leaves are utilized as a green vegetable, but the primary output is the calyx, which is utilized in beverages [3]. Therefore, roselle has been exploited for use in the food, cosmetic, and pharmaceutical industries [4]. Its fruits and flowers may be used to treat bronchitis and cough [5]. Such attributes are due to the presence of variable amounts of multiple beneficial ingredients in roselle, such as minerals, vitamin C, carotene, amino acids, organic acids, and sugars in the calyces [6], as well as anthocyanins, flavonoids, steroids, triterpenoids, alkaloids [7], valuable micronutrients, calories, protein, fiber [8], and fixed oils (17%) [9]. Many of these are powerful antioxidants which inhibit α-glucosidase, α-amylase, and angiotensin-converting enzymes, thereby modulating the calcium pathway. Phenolic acids (particularly protocatechuic acid), organic hydroxy citric acid, hibiscus acid, and anthocyanins (dyl-phenidin-3-sambobioside and cyanidin-3-sambobioside) achieve these beneficial effects [4,7].

Throughout their entire life cycles, plants are exposed to varied stressful agents. One of the most harmful stresses limiting agricultural output globally is salinity [10]. Salinity may cause ion toxicity and osmotic stress in plants, destroying lipids, proteins, and DNA by creating reactive oxygen species (ROS, consisting of superoxide radicals, hydrogen peroxide, hydroxyl anions, and single oxygen atoms) in plant cells [10,11].

Salt stress is a harmful factor limiting development and crop yield. About 23% of cultivated lands globally are subject to salinity constraints, and about 37% are sodic [11]. Soil salinity is defined as the existence of excess levels of soluble salts in the root zone, affecting soil moisture. This causes increased osmotic pressure, which affects plant growth and limits water absorption by plant roots [12], as well as interfering with important nutritive ion uptake [13]. Salinity can be initiated within the upper soil layer (30 cm) due to scarce or erratic rainfall, irrigation with low-quality water, water leakage from neighboring farmers, and surface runoff from high to low areas, hampering crop growth and development, and ultimately leading to physiological drought [14].

Lately, great attention has been paid to a growing interest in introducing natural biomaterials to increase plant tolerance to saline soils [15]. In foliar fertilization, plant biostimulants, hormones, nutrients, and other important materials are applied to plant leaves and stems during development. This method may enhance crop nutrient balance, accelerate yield and quality, increase disease resistance, and enhance drought tolerance [15].

Recently, foliar application of micronutrients and biostimulants has, in some cases, shown more effectiveness than soil application [16]. The use of the silicon compounds such as potassium silicate (KSi) as a foliar spray depends on determining the optimal quantities for the plant in case of non-available KSi in the soil, which leads to improved KSi absorption with positive effects [15].

Silicon is considered the second-most abundant mineral in the Earth’s crust [17], constituting 27.7% of the total weight in soil after oxygen content (47%) [18]. Its content comprises about 200–300 g kg^−1^ in clay soils and 450 g kg^−1^ in sandy soils [19], and varies from 1–45% in soil dry weight [20]. Moreover, KSi may increase plant photosynthesis, nutrient and water uptake, cell division, and pigment amounts [21], as well as sustain healthy plant development [22]. Hence, without KSi, plants may suffer subtle nutrient deficiencies. KSi deficiency leads to a decrease in photosynthesis, lower brix levels, increased wilting, higher infection prevalence and arthropod infestations, and decreased postharvest yield, all of which are stress signals [23].

KSi is a widely available source of potassium and silicon and is utilized in agricultural output mainly as a silica amendment with the added benefit of supplying a limited quantity of potassium. Potassium nutrition can contribute in several horticultural crops to enhance tree yields, fruit size, soluble solids, fruit color, ascorbic acid levels, shelf life, and shipping quality [24]. KSi does not include any volatile organic compounds, so its usage does not release hazardous or environmentally-persistent byproducts [25]. The use of KSi on some cereal species under water-deficient irrigation has produced the highest biomass yield responses [26].

Moreover, spraying a composite of KSi-nanomaterials results in a reduction in the adverse effects of drought stress on crops [27]. KSi shows many positive impacts on plants [28,29,30]: it (1) increases structural strength; (2) plays an active role in many physiological processes (e.g., regulating the uptake of other plant nutrients); and (3) improves plant growth and development, especially during abiotic (acidity, salinity, drought, etc.) and biotic stresses due to its ability to increase plant resistance through enhancing defense mechanism(s) [31]. Plants deprived of KSi are structurally weaker than KSi-enriched plants, resulting in a reduction in growth, viability, and reproduction and increased susceptibility to biotic and abiotic stresses [31].

Nowadays, actual, practical interest toward the use of aloe has been increasing. *Aloe vera* L. is a perennial, evergreen herb that thrives primarily in tropical and subtropical climates [32]. It contains large contents of polysaccharides, aloin, different essential minerals, amino acids, vitamins, and other active compounds [32]. Therefore, aloe species have been applied for thousands of years globally as a classic medicinal herb [33], as well as in cosmetic repair and health care [32].

Among the primary aloe species are (*Aloe vera* L.), (*Aloe maculata* L.), and (*Aloe saponaria* L.), which is often known as African aloe, soap aloe, or zebra aloe, and is an arid-zone plant endemic to eastern South Africa, Botswana, and Zimbabwe [34]. The juicy leaf sap makes suds in water and can be used as a soap substitute [34]. It is characterized by sharp spines on each leaf with white patches [35] and a short stem, and its rhizome elongation is characterized by fast spread and by germinating and growing in clumps. The big, thick, fleshy green leaves of *A. saponaria* have abundant gel with a high ornamental value [32]. *A.*
*saponaria* reportedly has actions of cardiac and immune reaction stimulation [36,37] and antinociceptive and anti-inflammatory activities [38]. *A. saponaria* mannan can inhibit the activation and proliferation of tumor cells [39]. Additionally, *A. saponaria* possesses antioxidant capacity, antiradical activity, and lipid peroxidation inactivation activity, and contains a maximum amount of phenols and flavonoids [34].

Therefore, the objective of the present study was to investigate the sole or dual interaction effects of applying a foliar spray of *A. saponaria* extract (Ae) or KSi on reducing the stressful salinity impacts on the development, yield, and features of roselle plants.

## 2. Materials and Methods

### 2.1. Field Trials

Two field trials during the summers of 2019 and 2020 were conducted at an experimental farm of the Department of Agriculture, Fayoum University, Fayoum, Egypt. Three experimental sites (S1, S2, and S3) were selected on the experimental farm with the following geographic coordinates: (1) S1 = (29°17′42′ N; 30°55′02′ E); (2) S2 = (29°17′41′ N; 30 55′04′ E); and (3) S3 = (29°17′45′ N; 30°55′06′ E). The soil was sandy clay. The average temperature during the experiment was 34.8 °C, the relative humidity was 64.8%, and the photoperiod was 13 h light and 11 h dark. Soil surface samples (30 cm) were collected to analyze their chemical and physical properties following the methodologies of Jackson [40] and Black et al. [41]. The soil analysis data are summarized in Table 1.

### 2.2. Experimental Design and Treatments

The experiment layout was a split-plot system based on a randomized complete block design with three replications. Treatments in the main plots were classified into three levels of salinity: (1) normal soil (ECe < 4 dS/m) (S1); (2) moderately-saline soil (ECe: 4–8 dS/m) (S2); and (3) highly-saline soil (ECe: 8–16 dS/m) (S3). In sub-plots, they were also classified into three KSi concentrations (0, 30, and 60 g L^−1^) (0, 12, and 24 L of K_2_SiO_3_ ha^−1^, respectively). In sub-sub-plots, they were further divided into three different application levels of Ae: 0% (0 cm^3^ L^−1^), 0.5% (5 cm^3^ L^−1^), and 1% (10 cm^3^ L^−1^). KSi and Ae were either applied as individual treatments or as a combined treatment. The experimental plots comprised an area of 5.76 m^2^ (2.4 m × 2.4 m) containing four edges 60 cm apart and six ridges 40 cm apart. At 20 days after sowing (DAS), the seedlings were thinned to a spacing of only one plant on each hill. The spray solutions were prepared in a 20 L atomizer containing drops of Triton B for wetting.

KSi and Ae extract were sprayed on plant leaves in the early morning in three stages: 45 days after vegetative growth, 75 days after stem elongation, and 105 days after flowering. All agricultural practices were applied according to the Egyptian Ministry of Agriculture recommendations. 

### 2.3. Preparation of Ae

Following the procedure recommended by Mabusela et al. [42], three-year-old aloe leaves were cut from plants grown at the same experimental site, and were cold-pressed using a stainless-steel drum to obtain the aloe gel. Thereafter, aloe leaf tissues were crushed using a blender, and the extract was filter-sterilized. Distilled water was added to the aloe leaf tissues at an equal proportion (1:1 by volume); then the mixture was filtrated and kept at 4 °C, and subsequently used as the stock solution. Distilled water was added to create three concentrations of the obtained extract for foliar spray: (1) 0% (0 cm^3^ L^−1^); (2) 0.5% (5 cm^3^ L^−1^); and (3) 1% (10 cm^3^ L^−1^). Determination of phytohormones (gibberellins, GA_3_, indole-3-acetic acid (IAA), and abscisic acid, ABA) in Ae was carried out according to the method recommended by Nandi et al. [43]. Determination of minerals and sugars in Ae was carried out according to Rowe [44] (Table 2)**.**

### 2.4. Plant Sampling

Random samples (*n* = 9) were collected from each treatment at full blooming (145 DAS) to evaluate the growth and biochemical parameters and at 165 DAS to estimate yield traits and seed chemical compositions.

### 2.5. Morphological Features and Yield Attributes

On October of both seasons, the morphological characteristics of plants, plant height (cm), fresh wight (FW) (g), dry weight (DW) (g), and stem diameter (mm) were determined. Additionally, the number of fruit and fruit FW (g), sepal FW (g) and DW (g), and seed weight (g) were recorded at harvest for each plant.

### 2.6. Chlorophyll and Performance Index Measurements

Chlorophyll content was determined following the method recommended by Arnon [45]. A clean mortar and pestle, acetone (10 mL at 80% *v/v* for each sample), and 0.2 g of fresh tissue was used to extract chlorophyll. After filtration, the optical densities of the filtrates (supernatants) were monitored at 663, 645, and 480 nm with a UV-120 spectrophotometer (Shimadzu Corp., Kyoto, Japan).

In the field, chlorophyll fluorescence was measured in the fresh leaves using a fluorometer (FMS-2, portable, pulse-modulated, Hansatech, Norfolk, UK). The upper fourth leaf on each plant was subjected to light for 2 min until a constant rate of photosynthesis was reached. Steady-state fluorescence (Fs), maximum light-adaptive fluorescence (Fm), variable fluorescence (Fv), and minimum-adaptive fluorescence (F0) were measured [46]. Fv/Fm (maximum quantum yield of photosystem II, PSII) was calculated [47].

The performance index (PI) of photosynthesis was estimated based on equal absorption (PIABS) following Clark et al. [48]. Additionally, single-photon avalanche diode (SPAD) was measured by a portable chlorophyll meter (SPAD-502, Konica Minolta Sensing, Inc., Osaka, Japan).

### 2.7. Determinations of Relative Water Content (RWC %) and Membrane Stability Index (MSI %)

The RWC was evaluated according to Hayat et al. [49] using 20 disks of 2 cm diameter from midrib-free fresh upper leaves. The fresh mass (FM) and turgid mass (TM) of the disks were recorded by weighing and then transferring them to a dark location to be saturated by immersion in completely ion-free distilled water for 24 h. After dry-blotting any water, TM was measured. The disks’ dry mass (DM) was also measured after dehydrating in an electric oven. To calculate the RWC, the following formula was applied:RWC (%) = [(FM − DM)/(TM − DM)] × 100 

The MSI was evaluated using the method recommended by Premachandra et al. [50]. Two samples (0.2 g each) were taken from fresh leaf tissues. Both samples were heated at 40 °C for 30 min and boiled at 100 °C for 10 min after being immersed in test tubes with 10 mL of completely-deionized distilled water. Using a conductivity bridge (Starlac Industries, Ambala, Haryana, India), the solution electrical conductivity was recorded for both solutions (EC1 and EC2). To calculate the MSI, the following formula was applied:MSI (%) = [1 − (EC1/EC2)] × 100 

### 2.8. Determination of Nutrients

A known weight of 0.1 g dry tissue of the different leaf samples was extracted with 25 mL of 80% ethanol for 24 h at room temperature. After filtration, the ethanol was removed by boiling the extract in a water bath; then, the extract was measured to a known volume (100 mL in a measuring flask). This extract was used to estimate the quantity of total soluble sugars according to Dubois et al. [51], total free amino acids according to Jayaraman [52], and leaf proline content by the rapid colorimetric method following Bates et al. [53]. One hundred mg of leaf powder was digested in sulfuric and perchloric acids for the determination of sodium and potassium, as outlined by Piper [54]. The concentration of sodium and potassium (mg kg^−1^ DW) was assessed by flame photometer (Gallenkamp Co., Cambridge, UK) [55].

Dried leaves were milled at 70 °C for three days; then 10 mg of dry matter was digested with 1.5 mL of a mixture of 1 M hydrochloric acid and 2.3 M hydrogen fluoride according to Novozamsky et al. [56]. The samples were placed on a shaker overnight and diluted to 1:10 (100 µL of sample and 900 µL of distilled water). Subsequently, 250 µL of boric acid (3.2%) was added to 50 µL of the sample and shaken overnight. Then, 50 µL of sample was added to 250 µL of tartaric acid and 250 µL of ascorbic acid. The KSi concentration in the leaves was assessed by the colorimetric molybdenum blue method at 811 nm with a spectrophotometer (Shimadzu Corp., Kyoto, Japan) as outlined by Van der Vorm [57].

Chloride was determined from ashed samples using hot water, which were then titrated with a standard silver nitrate solution [58]. Ash was estimated gravimetrically following Sluiter et al. [59].

### 2.9. Estimation of Physicochemical Characteristics: Sepal Anthocyanin, pH, Total Soluble Solids, and Acidity in Dry Roselle Calyces

Fifteen grams of calyces in dry matter basis were crushed in a mill to obtain aqueous extracts from each replicate of each treatment. Aliquots (2.5 g) of powder were added to 200 mL of distilled water in 250 mL glass bottles. The extracts were placed in a water bath at 40 °C for 15 min after standing for 24 h at room temperature in the dark.

#### 2.9.1. pH, Total Soluble Solids

With a Bellingham Stanley LTD digital hand refractometer Model (Opti Brix 85), titratable acidity was assessed for the percentage of citric acid present. The measurement procedures followed those set by the Association of Official Analytical Chemists (AOAC) [60].

#### 2.9.2. Total Anthocyanin

Anthocyanin concentration was assessed following Abdel-Aal and Hucl [61]. After weighing 100 mg of each crushed sample into centrifuge tubes, 10 mL of acidified methanol extractor solution (methanol: 1.5 N HCl at a ratio of 85:15 *v/v*) was added. The combination was left to stand for 24 h at 4 °C in the dark. The extractor solution was then filled to the primary volume to avoid evaporation-related variations. In a Sigma centrifuge, the samples were spun at 5000 rpm for 20 min. A 1:10 dilution was created (100 L of pure extract + 900 L of acidified methanol). Finally, the absorbance of the diluted extract was measured in a spectrophotometer (Shimadzu) at 533 nm. The total anthocyanin content of calyces was measured in mg of anthocyanins per g of dried matter.

### 2.10. Statistical Analysis

All data were analyzed using analysis of variance (ANOVA) at a 5% probability level using InfoStat software version 2016, Facultad de Ciencias Agropecuarias, Universidad Nacioal de Córdoba, Argentina for a split-plot system in a randomized complete block design, after testing for homogeneity of error variances following the procedure outlined by Gomez [62]. The significant differences between treatments were compared using Duncan’s multiple range test [63].

## 3. Results

### 3.1. Effect of Salinity 

The highest rate of salinity adversely led to a remarkable decline in all growth and yield structures under investigation (plant height, stem diameter, plant fresh and dry weight and fruit fresh weight, fruit number, sepal dry weight, seed weight, and fruit ovary dry weight) (Table 3 and Table 4). These parameters significantly increased at the lowest level of salinity (S1) (Table 3 and Table 4).

In addition, the lowest level of salinity gave the highest value of chlorophyll fluorescence (Fv/Fm), photosynthetic PI, SPAD, RWC %, and MSI % compared with the other two high salinity levels (Table 5). Moreover, total soluble sugars, total free amino acids, proline, potassium, SiO_2_ (Table 6), anthocyanins, total soluble solids, and acidity (citric acid %) (Table 7) greatly increased with the lowest level of salinity, whereas pH, sodium, and chloride increased with highest level of salinity.

### 3.2. Effect of KSi

Spraying the plants with KSi at 30 g L^−1^ helped to significantly increase plant height, fruit fresh weight, fruit number, sepal dry weight, second-season yield (Table 3 and Table 4), chlorophyll fluorescence (Fv/Fm), photosynthetic PI, SPAD, RWC %, and MSI % (Table 5) over the control. Conversely, stem diameter, plant fresh and dry weight, seed weight, fruit ovary dry weight, sepal dry weight, and first-season yield (Table 3 and Table 4) increased with the application of 60 g L^−1^ KSi. Moreover, KSi at 60 g L^−1^ yielded the maximum values of total soluble sugars, total free amino acids, proline, potassium, SiO_2_ (Table 6), anthocyanins, total soluble solids, and acidity (citric acid %) (Table 7) and yielded the lowest pH values (Table 7) and sodium and chloride content (Table 6).

### 3.3. Effect of Ae

The Ae foliar spray had a significantly positive effect on all surveyed growth characteristics over the control treatment. The highest values of stem diameter and plant fresh and dry weight (Table 3), fruit fresh weight, fruit number (Table 4), SPAD chlorophyll reading (Table 5), total soluble sugars, total free amino acids, potassium (Table 6), and anthocyanins (Table 7) were recorded for a 0.5% Ae concentration. However, the highest plant height (Table 3), chlorophyll fluorescence (Fv/Fm), PI, RWC, MSI (Table 5), proline, SiO_2_ (Table 6), and total soluble sugars (Table 7) were obtained with the highest concentration of Ae (1%). Moreover, a 0% Ae concentration yielded the highest values of sepal dry weight, yield, seed weight, and fruit ovary dry weight (Table 4), as well as highest sodium and chloride content (Table 6) and pH level (Table 7).

### 3.4. Effect of Salinity Plus KSi

Spraying the plants with KSi at 30 g L^−1^ at the lowest level of salinity (S1) significantly increased plant height, fruit fresh weight, fruit number (Table 3 and Table 4), chlorophyll fluorescence (Fv/Fm), PI, SPAD chlorophyll reading, RWC %, MSI % (Table 5), sepal dry weight, and second-season yield (Table 4). On the other hand, stem diameter, plant fresh and dry weight, seed weight, fruit ovary dry weight, sepal dry weight, and first-season yield (Table 3 and Table 4), as well as total soluble sugars, total free amino acids, proline, K, SiO_2_ (Table 6), anthocyanins, total soluble solids, and acidity (citric acid %) (Table 7) increased with the application of 60 g L^−1^ at the lowest level of salinity (S1). The maximum pH (Table 7) and sodium and chloride contents were recorded at the highest level of salinity (S3) with 0 g L^−1^ of KSi (Table 6).

### 3.5. Effect of Salinity Plus Ae

The highest values of stem diameter, plant fresh and dry weight (Table 3), fruit fresh weight, fruit number (Table 4), SPAD chlorophyll reading (Table 5), total soluble sugars, total free amino acids, K (Table 6), and anthocyanins (Table 7) were recorded at an Ae concentration of 0.5% at the first level of salinity. The highest plant height (Table 3), chlorophyll fluorescence (Fv/Fm), PI, RWC, MSI (Table 5), proline, SiO_2_ (Table 6), and total soluble sugars (Table 7) levels were obtained with the highest Ae concentration (1%) along with the lowest salinity level (S1). In contrast, Ae concentration at 0% with the lowest salinity level (S1) yielded the highest values of sepal dry weight and yield, seed weight, and fruit ovary dry weight (Table 4). The highest values of sodium and chloride (Table 6), and pH values (Table 7) were recorded for 0% Ae concentration at the highest salinity level (S3).

### 3.6. Effect of Interaction of Ae Plus KSi

The combination between Ae either at 1% with KSi at 30 g L^−1^ yielded the highest plant height (Table 3), chlorophyll fluorescence (Fv/Fm), PI, RWC, and MSI (Table 5). On the other hand, the interaction between a 0.5% Ae concentration plus KSi at 60 g L^−1^ greatly maximized stem diameter, plant fresh and dry weight (Table 3), total soluble sugars, total free amino acids, potassium (Table 6), and anthocyanins.

Plants with 0.5% Ae plus KSi at 30 g L^−1^ yielded the highest values of fruit number, fruit fresh weight (Table 4), and SPAD chlorophyll reading (Table 5). Moreover, the application of Ae at 1% plus KSi at 60 g L^−1^ produced the maximum values of proline, SiO_2_ (Table 6), total soluble sugars, and acidity, while plants treated with both Ae and KSi had the highest sodium and chloride concentrations (Table 6) and pH values (Table 7).

Ae concentration at 0% with either 30 or 60 g L^−1^ of KSi yielded the highest sepal dry weight and yield, respectively, in both seasons (Table 4).

### 3.7. Combination of Salinity Plus KSi and Ae

The plants which were sprayed with Ae at 1% along with the low concentration of KSi at the first salinity level (S1) yielded the highest plant height, chlorophyll fluorescence (Fv/Fm), PI, RWC, and MSI (Table 3 and Table 5). The interaction between Ae at 0.5% plus KSi at 60 g L^−1^ and the lowest salinity level (S1) greatly maximized stem diameter, plant fresh and dry weight (Table 3), total soluble sugars, total free amino acids, potassium (Table 6), and anthocyanins (Table 7).

The highest values of fruit number, fruit fresh weight, and SPAD were observed from plants that had Ae at 0.5% plus KSi at 30 g L^−1^ with the lowest salinity level (S1) (Table 4 and Table 5).

On the other hand, proline, SiO_2_, total soluble sugars, and acidity reached their highest content levels with the interaction between Ae at 1% plus KSi at 60 g L^−1^ along with S1. Nevertheless, the highest sodium and chloride content and pH values were produced by the plants that received 0% of both Ae and KSi during the highest level of salinity (Table 6 and Table 7).

Ae concentration at 0% with 30 or 60 g L^−1^ of KSi along with S1 resulted in the highest values of sepal dry weight and yield, respectively, in both seasons (Table 4).

## 4. Discussion

Our results confirmed that salinity treatments sharply decreased all investigated parameters of growth, as shown in Table 3. Salinity stress negatively affected plant yield (Table 4), and in turn, their physiological traits, biochemical parameters, nutrient content, fruit yield, and fruit quality (Table 5, Table 6 and Table 7). Salinity stress is a complex phenomenon that causes osmotic stress, specific ion impacts, and deprivation of nutritive materials, all of which impact many physiological and biochemical pathways in plant development [64]. Tester and Bacic [13] also stated that high salt concentrations as a result of disarrangement in irrigated agriculture or natural processes causes inhibition of plant development and yield. This is because high salinity rates lead to ion imbalance due to drought stress and increased toxic levels of cytoplasmic sodium [65].

In plants, the ion toxicity stress caused by salinity is not a consequence of osmotic stress but rather due to the high cytoplasmic concentration of sodium. The osmotic stress is a consequence of the high salt concentration. Excess salt in the soil can affect plant growth through osmotic obstruction of root water intake, specific ion effects that might cause direct toxicity, and the insolubility or competitive uptake of ions, which can disturb plant nutrient balance [66]. A rivalry between sodium and potassium has been discovered, resulting in a lower level of internal potassium at high external sodium chloride concentrations [67,68]. This could affect potassium uptake by sodium chloride, resulting in potassium deficiency, and increase the sodium/potassium ratio, reducing plant development and causing ionic poisoning [67,69]. 

Increased sodium chloride levels may cause nutritional disruptions, reducing plant development by altering nutrient availability, transportation, and partitioning [12]. The drop in potassium level in plants exposed to salinity demonstrated this clearly. Under salinity stress, the level of potassium and other nutrients was also reduced in fennel [70], *Trachyspermum ammi* [71], peppermint and lemon verbena [72], and *Matricaria recutiton* [72]. Due to the chemical similarity between sodium and potassium ions, sodium has a strong inactivation impact on potassium absorption. Other research has found that people who are exposed to salt accumulate more sodium and chloride, which may be linked to osmotic or water potential, as well as ionic toxicity [73,74]. 

As a result of ionic imbalance, salinity has been shown to have a possible deleterious effect on the internal water state of herbs [75,76] that can disturb leaf RWC, stomatal conductance, and osmotic and turgor potential [74]. This may lead to lower plant vigor in many species [77], and is supported by the loss of water content from other *Hibiscus* species in saline environments [78].

Moosavi et al. [79] concluded that salinity stress markedly lowered development parameters of *H.*
*sabdariffa* L. seedlings. The deleterious effects of salinity were also recorded by Mohamed et al. [74] on the physiological and biochemical properties of roselle plants, as well as by Parida et al. [80] on *Bruguiera parviflora*. Akram et al. [81] stated that in sunflower, 150 mM sodium chloride markedly lowered the quantum yield of PSII measured as Fv/Fm. Ghabour et al. [82] have recorded that the development and yield of roselle plants also decreased in elevated soil salinity.

The effects of salt stress on photosynthetic enzyme activities could be a secondary effect mediated by the lower CO_2_ partial pressure in the leaves caused by stomatal closure [83,84]. In this regard, Desingh and Kanagaraj [85] stated that salt stress could alter photosynthetic biochemistry by causing disorientation of the chloroplast lamellar system and loss of chloroplast integrity, resulting in a reduction in photosystem activity. Plants are vulnerable to oxidative stress because they generate ROS such as O_2_, H_2_O_2_, and OH when exposed to salt [86,87]. ROS generation may deleteriously impact plant cellular membrane integrity, enzyme activity, and the photosynthetic system [88].

When salt stress occurs, abscisic acid is produced, which seals stomata when delivered to guard cells. This in turn leads to photo-inhibition and photosynthesis decline, oxidative stress occurrence, and inhibition of cell expansion [12]. The reduction of chlorophyll in stressed plants could be due to the ineffectiveness of the thylakoid membrane; chlorophyll is degraded by the formation of proteolytic enzymes such as chlorophyllase, thereby destroying the photosynthetic mechanism [89]. This, in turn, can lower the plant photosynthetic rate [90] and inactivate ion accumulations [91].

Dolatabadian and Jouneghani [92] found that salinity stress causes an increase in free radicals in chloroplasts and the breakdown of chlorophyll molecules by ROS, resulting in decreased photosynthesis and development of the common bean. Proline is one of the most significant amino acids produced by plants, and accumulates in response to salinity [93]. Proline is an osmolyte and antioxidant that may aid plant survival by preserving cell turgor [94]. It is a protein amino acid with high structural rigidity required for primary metabolism. Proline has the potential to protect protein integrity while enhancing the activity of several enzymes [95,96]. Proline has been shown to improve plant growth and physiological, biochemical, and morphological properties, as well as provide antioxidant system protection during salinity exposure [97,98].

In contrast to the negative impact of salinity, the application of KSi at 30 or 60 g L^−1^ in tandem with different levels of salinity had a positive impact on improving the studied growth, yield characteristics, and nutrient uptake of roselle plants. Chlorophyll fluorescence and photosynthetic PI also increased with KSi application, which may reflect plant health status, and is also related to plant water availability and nutrition level [98]. Silicates, such as KSi, have a precious role in inducing plant resistance to abiotic stresses [99]. It could also help plants improve cell membrane integrity and stability under biotic or abiotic stress [100]. KSi releases silicon, which may reduce different biological and nonbiological stresses in plants by preserving the water potential in plants and increasing light activity and stomatal conductance of leaves under high rates of transpiration [101,102,103]. In addition, KSi improves the architecture for more erect leaves to intercept higher solar luminosity, and increases photosynthetic efficiency and chlorophyll content [104].

In addition, KSi may encourage cell division and the antioxidant defense system [105] by biosynthesizing carbohydrates, activating certain enzymes such as glutathione peroxidase, and minimizing ROS, thereby preventing plant aging and death [105]. Furthermore, potassium plays a significant role in plant nutrition and can enhance the translocation of assimilates and protein synthesis [106], and regulates several plant metabolic processes. Therefore, it participates in cell enlargement and division and in increasing plant stress tolerance [107]. Moreover, spraying with KSi at 30 g L^−1^ alleviated the salinity stress, as indicated by low proline concentration with the addition of KSi. This may be an indicator of stress tolerance, as the concentration of proline was much higher with the control treatment (0 g L^− 1^) but was greatly reduced with KSi application. 

One mechanism of KSi accumulation that may explain this apparent stress tolerance may be the ability to form a thick silicated layer on the leaf surface that effectively reduces cuticular transpiration [108] through the physical block created by the deposition of this element under the cuticle and on the epidermal cell wall. Another mechanism may be the enhancement of defense mechanisms such as the production of phenolic compounds, thereby increasing lignification and promoting cell wall strengthening for controlling several stress factors in plants [109]. Shaaban and Abou El-Nour [110] have suggested that foliar application of KSi could be helpful toward silica deposition to enable plants to maintain healthy hairy roots to better absorb water and macro- and micronutrients. As a result, KSi could partially alleviate salt stress by disabling oxidative membrane damage, and also influence plant osmotic adjustment [111].

In addition, potassium helps to promote photosynthesis, and activates enzymes and coenzymes to metabolize carbohydrates for the manufacture of starch and protein. Thus, this could explain the increase of carbohydrates content through foliar spraying with KSi.

Additionally, the potassium content gradually increased as KSi concentration increased, which may account for the increase in growth attributes with KSi foliar spray; potassium is the most abundant cellular cation, and has a critical role in maintaining cell turgor, enzyme activities, and membrane potential.

Aloe juice has several similar water-soluble polymer materials, which may greatly help to increase the absorption of the effective ingredient of mucopolysaccharide in aloe leaves, thereby accelerating cell division and promoting cell metabolism [32]. Such an effect could be more beneficial for clay soils, which have higher salinity. Our results clearly observed the decrease in the sodium and chloride concentration in roselle plants, especially with a low level of Ae application. Ae improved the quality of roselle plants by increasing the anthocyanin and nutrient content of potassium, and KSi also reduced the pH of the extract. Ae significantly enhanced the activity of proline to help plants deal with salt stress, which may be due to the naturally-occurring antioxidant components found in the aqueous extract of aloe leaves (e.g., total phenols, flavonoids, ascorbic acid, β-carotene, and α-tocopherol) [112]. *A.*
*saponaria* also contains phenols and flavonoids which could contribute toward plant tolerance of deficit irrigation. Jain et al. [34] reported that *A. saponaria* water extract has the highest antioxidant activity against lipid peroxidation, suggesting the contribution of polar phytoconstituents in inhibiting lipid peroxidation [113].

Moreover, amino acids play a role in the alternative routes of IAA synthesis and plant growth and development through their influence on gibberellins [114]. Hence, Ae greatly decreased the osmotic effects of salt on plant growth yield, and the photosynthesis process of roselle plants. Salinity caused a significant fall in the actual (photochemical) efficiency of the photosystem and electron-transport chain, as well as a decrease in the abovementioned systems, as noted by Stepien and Klobus [115], which may cause the decrease in all growth characteristics and, consequently, in yield (Table 4). However, especially at the highest concentration, Ae application significantly enhanced the photosynthetic parameters, thereby producing better growth and yield. In addition, the minerals copper, zinc, manganese and iron [34] found in *A. saponaria* gel are needed for the proper function of enzyme production pathways and metabolism function. Therefore, a lack of minerals impacts the survivability of plants. Minerals constitute an essential part of various antioxidant enzymes and are needed to improve plant growth and development and for the plant to complete its life cycle [116], thereby enhancing plant tolerance to salinity.

## 5. Conclusions

KSi and Ae can be applied as natural nutrients to cultivate roselle plants under salinity conditions. Such nutrients could be sprayed at 30 g L^−1^ for KSi or 0.5% (5 cm^3^ L^−1^) for Ae to improve plant characteristics of growth, yield, quality, and physiological attributes, reducing the negative impacts of salinity by stimulating nutrient balance, cell integrity, and chemical composition inside the plant.

## Figures and Tables

**Table 1 plants-11-00497-t001:** Initial soil physicochemical properties.

Soil Salinity Level	Depth (cm)	Particle Size Distribution			Texture Class	ρ_b_ g cm^−3^	K_sat_ cm h^−1^	FC %	WP %	AW %	ECe (dS/m)	pH	OM %	CEC	CaCO_3_ %
		Sand %	Silt %	Clay %											
Normal soil (Ece < 4 dS/m)	0.0–20	80	9.5	10.5	SL	1.57	1.87	23.29	10.42	12.87	3.56	7.40	0.84	11.89	6.58
	20–40	75.5	12.3	12.2	SL	1.53	1.67	22.61	10.33	12.28	3.44	7.61	0.81	10.40	7.21
	40–60	73.3	13.2	13.5	SL	1.50	1.66	21.76	11.22	10.54	3.72	7.38	0.75	10.13	6.09
	mean	76.27	11.67	12.07	SL	1.53	1.73	22.55	10.66	11.90	3.57	7.46	0.80	10.81	6.63
Moderately saline soil (Ece: 4–8 dS/m)	0.0–20	78.77	10.30	10.93	SL	1.58	2.11	20.38	10.61	9.77	6.82	7.55	0.79	10.67	7.11
	20–40	76.17	11.21	12.63	SL	1.53	1.94	21.75	10.87	10.88	7.22	7.63	0.73	9.84	7.98
	40–60	74.21	13.15	12.64	SL	1.55	1.89	22.41	11.81	10.60	7.35	7.25	0.64	9.21	6.18
	mean	76.38	11.55	12.06	SL	1.55	1.98	21.51	11.10	10.42	7.13	7.48	0.72	9.91	7.09
Highly saline soil (Ece: 8–16 dS/m)	0.0–20	75.23	10.47	14.30	SL	1.49	1.67	22.20	11.79	10.41	10.87	7.37	0.81	11.38	6.44
	20–40	77.17	12.76	10.07	SL	1.54	2.18	19.33	10.42	8.91	11.22	7.74	0.63	9.14	7.21
	40–60	76.13	11.66	12.21	SL	1.51	1.84	20.63	11.15	9.48	11.07	7.66	0.62	9.99	8.16
	mean	76.18	11.63	12.19	SL	1.51	1.90	20.72	11.12	9.60	11.05	7.59	0.69	10.17	7.27

SL, Sandy loam; ρ_b_, bulk density; K_sat_, saturated hydraulic conductivity; FC, field capacity; WP, wilting point; AW, available water; ECe, electrical conductivity; OM, organic matter; CEC, cation exchangeable capacity.

**Table 2 plants-11-00497-t002:** Determination of phytohormones (gibberellins, GA_3_, indole-3-acetic acid (IAA), and abscisic acid, ABA), minerals, and sugars in aloe extract.

Parameters		Minerals		Sugars		Sugars	
GA_3_ mg 100 g^−1^	15.00	N mg 100 g^−1^	82.65	Glucuronic (%)	2.01	Rhamnose (%)	0.08
IAA mg 100 g^−1^	0.63	P mg 100 g^−1^	7.95	Stachyose (%)	2.48	Mannose (%)	0.10
ABA mg 100 g^−1^	3.06	K mg 100 g^−1^	57.14	Galacturonic (%)	1.68	Raffinose (%)	0.40
Carbohydrates %	8.70	Fe ppm	766.11	Sucrose (%)	0.30	Arabinose (%)	0.24
Vitamin C mg g^−1^	154.64	Zn ppm	166.87	Maltose (%)	2.54	Fructose (%)	0.45
Protein %	3.70	Mn ppm	478.88	Lactose (%)	0.22	Mannitol (%)	0.06
Cholesterol mg g^−1^	18.73	Ca mg 100 g^−1^	37.00	D-glucose (%)	0.32	Sorbitol (%)	0.02
Polyphenol’s µg g^−1^	22.82	Cu ppm	42.73	Glucose (%)	0.64	Ribose (%)	0.12
Flavonoid µg g^−1^	2.28	Mg mg 100 g^−1^	15.55	Xylose (%)	0.15	Total sugars in ppm	12.05
Total sterol µg g^−1^	65.47	Na mg 100 g^−1^	48.27	Galactose (%)	0.24	Total sugars in %	0.001
Polysaccharides %	90						
Antioxidant activity %	47.1						

**Table 3 plants-11-00497-t003:** Effect of foliar spraying with aloe extract (Ae), potassium silicate (KSi) under different salinity levels and their interaction on vegetative growth of roselle plants.

Salinity Levels	Ae %	KSi g L^−1^	Plant Height (cm)		Stem Diameter (mm)		Fresh Weight Plant^−1^ (kg)		Dry Weight Plant^−1^ (g)	
			S-I	S-II	S-I	S-II	S-I	S-II	S-I	S-II
S1 (ECe < 4 dS/m)	0	0	213.67 ^abc^	212.00 ^abcd^	26.62 ^cd^	27.49 ^bcd^	1.90 ^cd^	1.33 ^c^	531.58 ^bc^	389.21 ^d^
		30	217.33 ^ab^	219.33 ^abc^	26.73 ^cd^	29.36 ^abcd^	2.33 ^ab^	1.47 ^c^	653.52 ^a^	404.33 ^d^
		60	226.67 ^a^	236.00 ^a^	28.33 ^bc^	33.14 ^abc^	2.12 ^ab^	1.57 ^bc^	648.54 ^a^	541.87 ^bc^
	0.5	0	212.33 ^abc^	218.00 ^abc^	26.23 ^cde^	27.58 ^bcd^	2.05 ^bc^	1.50 ^c^	609.69 ^ab^	462.71 ^cd^
		30	238.00 ^a^	234.33 ^a^	28.07 ^bcd^	29.70 ^abcd^	1.93 ^bc^	1.97 ^ab^	605.79 ^ab^	628.41 ^ab^
		60	219.33 ^ab^	221.67 ^ab^	35.95 ^a^	35.09 ^a^	2.50 ^a^	2.33 ^a^	709.13 ^a^	736.92 ^a^
	1	0	230.00 ^a^	230.67 ^a^	30.67 ^abc^	26.72 ^cde^	1.47 ^de^	1.93 ^ab^	447.16 ^cd^	610.71 ^ab^
		30	236.67 ^a^	240.33 ^a^	32.80 ^ab^	34.68 ^ab^	1.20 ^de^	2.30 ^a^	379.05 ^d^	670.81 ^ab^
		60	234.67 ^a^	238.33 ^a^	31.91 ^abc^	32.16 ^abc^	1.17 ^g^	2.15 ^a^	399.57 ^d^	665.77 ^ab^
S2 (ECe 4–8 dS/m)	0	0	167.00 ^efg^	170.33 ^efgh^	13.18 ^i^	14.80 ^g^	0.19 ^ef^	0.23 ^ef^	72.12 ^g^	72.57 ^fg^
		30	179.00 ^de^	179.67 ^ef^	20.67 ^efg^	18.69 ^fg^	0.53 ^g^	0.48 ^def^	128.59 ^efg^	124.96 ^efg^
		60	182.00 ^de^	187.67 ^de^	15.08 ^ghi^	15.41 ^fg^	0.62 ^fg^	0.27 ^ef^	101.70 ^fg^	106.74 ^efg^
	0.5	0	176.67 ^def^	183.67 ^de^	16.56 ^fghi^	18.13 ^fg^	0.59 ^fg^	0.38 ^def^	113.29 ^efg^	110.65 ^efg^
		30	192.67 ^bcde^	196.67 ^bcde^	18.60 ^fghi^	18.84 ^fg^	0.85 ^fg^	0.80 ^d^	222.54 ^e^	225.92 ^e^
		60	186.67 ^cde^	190.00 ^cde^	22.29 ^def^	22.54 ^def^	0.33 ^g^	0.56 ^de^	189.94 ^ef^	189.61 ^ef^
	1	0	166.00 ^efg^	170.67 ^efg^	16.66 ^fghi^	14.90 ^g^	0.33 ^g^	0.29 ^ef^	98.04 ^fg^	98.83 ^efg^
		30	196.67 ^bcd^	196.67 ^bcde^	19.08 ^fgh^	20.06 ^efg^	0.31 ^g^	0.60 ^de^	122.22 ^efg^	122.08 ^efg^
		60	187.33 ^cde^	184.00 ^de^	18.06 ^fghi^	15.09 ^g^	0.31 ^g^	0.34 ^ef^	108.83 ^fg^	107.15 ^efg^
S3 (ECe 8–12 dS/m)	0	0	117.00 ^i^	117.67 ^k^	12.79 ^i^	12.95 ^g^	0.15 ^g^	0.14 ^f^	44.44 ^g^	44.33 ^g^
		30	141.33 ^ghi^	146.33 ^ghijk^	16.15 ^ghi^	16.02 ^fg^	0.23 ^ef^	0.23 ^ef^	70.67 ^g^	73.94 ^fg^
		60	123.33 ^hi^	120.00 ^jk^	16.24 ^ghi^	13.92 ^g^	0.26 ^g^	0.25 ^g^	63.07 ^g^	64.16 ^fg^
	0.5	0	122.67 ^hi^	140.33 ^hijk^	13.52 ^hi^	12.90 ^g^	0.19 ^g^	0.19 ^ef^	83.67 ^fg^	82.95 ^fg^
		30	147.00 ^gh^	149.33 ^ghij^	17.33 ^fghi^	15.59 ^fg^	0.26 ^g^	0.25 ^ef^	99.80 ^fg^	100.89 ^efg^
		60	140.00 ^ghi^	142.33 ^ghijk^	16.87 ^fghi^	15.98 ^fg^	0.27 ^g^	0.26 ^ef^	86.34 ^fg^	88.32 ^fg^
	1	0	125.33 ^hi^	118.33 ^k^	14.79 ^hi^	14.38 ^g^	0.20 ^g^	0.21 ^ef^	61.60 ^g^	62.68 ^fg^
		30	135.67 ^hi^	131.00 ^ijk^	18.28 ^fghi^	16.82 ^fg^	0.37 ^g^	0.36 ^ef^	90.00 ^fg^	90.88 ^fg^
		60	149.67 ^fgh^	150.67 ^fghi^	15.53 ^ghi^	14.65 ^g^	0.31 ^g^	0.33 ^er^	81.30 ^fg^	82.90 ^fg^
LSD 5%	(S)		12.14	11.68	2.10	2.02	0.13	0.28	39.96	86.78
	(A)		13.37	10.93	2.62	1.51	0.23	0.15	37.32	66.53
	(KSi)		9.09	10.02	1.94	2.41	0.40	0.25	37.05	43.83
	(S*A)		23.16	18.92	4.54	2.61	0.16	0.14	64.65	115.24
	(S*KSi)		15.75	17.35	3.36	4.17	0.27	0.24	64.18	75.91
	(A*KSi)		15.75	17.35	3.36	4.17	0.27	0.24	64.18	75.91
	(S*A*KSi)		27.27	30.05	5.82	7.23	0.47	0.42	111.16	131.48

S-I: 1st season, S-II: 2nd season. Values with the same letter within a column for each treatment are not significantly (*p* > 0.05) different, according to Duncan’s multiple range test.

**Table 4 plants-11-00497-t004:** Effect of foliar spraying with aloe extract (Ae), potassium silicate (KSi) under different salinity levels and their interaction on yield attributes of roselle plants.

Salinity Levels	Ae %	KSi g L^−1^	No of Fruits Plant^−1^		Fruits Fresh Weight Plant^−1^ (g)		Sepals Weight Plant^−1^ (g)		Fruits Ovaries Weight Plant^−1^ (g)		Seeds Weight Plant^−1^ (g)		Sepals Yield Dry Weight t h^−1^	
			S-I	S-II	S-I	S-II	S-I	S-II	S-I	S-II	S-I	S-II	S-I	S-II
S1 (ECe < 4 dS/m)	0	0	31.00 ^fg^	35.42 ^f^	372.33 ^de^	330.78 ^d^	27.96 ^bc^	22.17 ^d^	80.83 ^c^	79.45 ^c^	25.97 ^cd^	28.86 ^c^	0.932 ^bc^	0.739 ^d^
		30	60.33 ^bcd^	60.14 ^bc^	396.33 ^de^	385.94 ^cd^	34.72 ^ab^	38.71 ^a^	119.41 ^a^	121.50 ^a^	38.69 ^ab^	40.59 ^a^	1.157 ^ab^	1.290 ^a^
		60	55.00 ^cd^	59.17 ^cd^	434.00 ^cd^	457.75 ^b^	37.88 ^a^	34.75 ^b^	132.33 ^a^	124.87 ^a^	47.72 ^a^	46.36 ^a^	1.263 ^a^	1.158 ^b^
	0.5	0	41.00 ^ef^	48.92 ^de^	435.67 ^cd^	428.72 ^bc^	16.87 ^d^	13.93 ^e^	67.66 ^cd^	68.15 ^de^	17.80 ^de^	14.77 ^de^	0.562 ^d^	0.464 ^e^
		30	78.33 ^a^	80.06 ^a^	687.33 ^a^	623.97 ^a^	36.78 ^a^	33.30 ^b^	102.50 ^b^	104.85 ^b^	39.02 ^ab^	43.97 ^a^	1.226 ^a^	1.110 ^b^
		60	67.67 ^ab^	70.03 ^ab^	538.00 ^bc^	574.28 ^a^	23.84 ^cd^	22.07 ^d^	82.13 ^c^	79.26 ^cd^	35.67 ^bc^	34.78 ^b^	0.795 ^cd^	0.736 ^d^
	1	0	38.33 ^f^	44.19 ^ef^	297.33 ^e^	243.44 ^e^	21.52 ^cd^	20.83 ^d^	60.07 ^d^	64.96 ^e^	16.81 ^def^	20.00 ^d^	0.717 ^cd^	0.694 ^d^
		30	63.33 ^bc^	65.61 ^bc^	598.00 ^ab^	613.00 ^a^	26.64 ^c^	26.14 ^c^	104.08 ^b^	102.26 ^b^	31.90 ^bc^	29.49 ^bc^	0.888 ^c^	0.871 ^c^
		60	51.00 ^de^	56.83 ^cd^	420.00 ^cde^	382.50 ^cd^	22.06 ^cd^	21.09 ^d^	77.83 ^c^	77.16 ^cd^	28.44 ^bcd^	27.32 ^c^	0.735 ^cd^	0.703 ^d^
S2 (ECe 4–8 dS/m)	0	0	8.00 ^i^	8.92 ^h^	62.00 ^f^	64.14 ^h^	3.58 ^e^	3.37 ^gh^	11.04 ^f^	10.40 ^h^	3.94 ^g^	4.70 ^gh^	0.119 ^e^	0.112 ^gh^
		30	9.00 ^hi^	9.25 ^h^	85.67 ^f^	85.81 ^gh^	3.98 ^e^	4.57 ^fgh^	12.32 ^f^	13.79 ^gh^	4.86 ^g^	5.37 ^fgh^	0.133 ^e^	0.152 ^fgh^
		60	11.00 ^hi^	9.67 ^h^	90.00 ^f^	95.56 ^fgh^	5.05 ^e^	5.16 ^fgh^	15.71 ^ef^	17.85 ^gh^	6.46 ^efg^	6.39 ^fgh^	0.168 ^e^	0.172 ^fgh^
	0.5	0	8.00 ^i^	7.33 ^h^	69.00 ^f^	69.08 ^h^	3.06 ^e^	3.03 ^h^	10.43 ^f^	11.63 ^h^	4.49 ^g^	5.18 ^fgh^	0.102 ^e^	0.101 ^h^
		30	16.00 ^hi^	15.06 ^gh^	136.33 ^f^	146.19 ^fg^	6.50 ^e^	6.64 ^fg^	20.31 ^ef^	21.08 ^fgh^	10.48 ^efg^	10.85 ^ef^	0.217 ^e^	0.221 ^fg^
		60	13.67 ^hi^	14.33 ^gh^	123.33 ^f^	124.03 ^fgh^	6.12 ^e^	6.13 ^fgh^	23.62 ^ef^	23.53 ^fg^	8.82 ^efg^	8.68 ^fg^	0.204 ^e^	0.204 ^fgh^
	1	0	9.67 ^hi^	8.53 ^h^	83.67 ^f^	81.17 ^gh^	3.65 ^e^	3.98 ^fgh^	10.82 ^f^	13.03 ^gh^	5.21 ^fg^	5.51 ^fgh^	0.122 ^e^	0.133 ^fgh^
		30	10.33 ^hi^	9.39 ^h^	91.67 ^f^	92.53 ^gh^	4.48 ^e^	4.82 ^fgh^	13.99 ^ef^	14.30 ^gh^	7.10 ^efg^	7.50 ^fgh^	0.149 ^e^	0.161 ^fgh^
		60	13.67 ^hi^	12.14 ^h^	92.67 ^f^	92.78 ^gh^	4.87 ^e^	4.91 ^fgh^	16.01 ^ef^	16.23 ^gh^	6.45 ^efg^	6.47 ^fgh^	0.162 ^e^	0.164 ^fgh^
S3 (ECe 8–12 dS/m)	0	0	7.00 ^i^	8.25 ^h^	71.33 ^f^	76.00 ^h^	3.64 ^e^	3.58 ^gh^	13.41 ^ef^	12.92 ^gh^	2.48 ^g^	2.91 ^gh^	0.121 ^e^	0.119 ^gh^
		30	14.67 ^hi^	11.89 ^h^	114.67 ^f^	106.47 ^fgh^	4.88 ^e^	4.86 ^fgh^	17.44 ^ef^	16.24 ^gh^	4.72 ^g^	4.59 ^gh^	0.163 ^e^	0.162 ^fgh^
		60	10.00 ^hi^	11.78 ^h^	91.00 ^f^	94.25 ^fgh^	3.94 ^e^	3.74 ^gh^	13.49 ^ef^	13.24 ^gh^	2.84 ^g^	2.85 ^h^	0.131 ^e^	0.125 ^gh^
	0.5	0	7.33 ^i^	8.33 ^h^	78.00 ^f^	77.72 ^h^	3.06 ^e^	3.01 ^h^	11.19 ^f^	12.01 ^h^	3.03 ^g^	3.08 ^gh^	0.102 ^e^	0.100 ^h^
		30	13.67 ^hi^	12.97 ^h^	103.00 ^f^	91.33 ^gh^	4.58 ^e^	4.82 ^fgh^	16.32 ^ef^	16.80 ^gh^	5.86 ^fg^	5.73 ^fgh^	0.153 ^e^	0.161 ^fgh^
		60	15.00 ^hi^	13.75 ^gh^	119.00 ^f^	111.08 ^fgh^	6.11 ^e^	6.09 ^fgh^	20.07 ^ef^	19.32 ^fgh^	4.52 ^g^	4.13 ^gh^	0.204 ^e^	0.203 ^fgh^
	1	0	11.67 ^hi^	14.64 ^gh^	113.67 ^f^	116.72 ^fgh^	5.29 ^e^	5.29 ^fgh^	20.71 ^ef^	20.04 ^fgh^	6.21 ^efg^	6.18 ^fgh^	0.176 ^e^	0.176 ^fgh^
		30	13.67 ^hi^	15.56 ^gh^	126.67 ^f^	125.97 ^fgh^	6.22 ^e^	6.04 ^fgh^	21.09 ^ef^	21.24 ^fgh^	7.14 ^efg^	7.46 ^fgh^	0.207 ^e^	0.201 ^fgh^
		60	19.67 ^gh^	23.56 ^g^	163.33 ^f^	159.44 ^f^	7.17 ^e^	7.36 ^f^	27.21 ^e^	29.41 ^f^	6.41 ^efg^	6.63 ^fgh^	0.239 ^e^	0.245 ^f^
LSD 5%	(S)		4.17	6.21	48.28	39.06	2.05	2.97	8.20	1.84	5.21	3.81	0.068	0.099
	(A)		6.64	1.96	62.30	10.00	1.95	1.45	5.37	4.72	4.32	1.89	0.065	0.048
	(KSi)		3.80	3.50	43.65	22.00	2.38	1.13	4.85	3.75	3.95	1.93	0.079	0.038
	(S*A)		11.49	3.40	107.91	17.32	3.38	2.5	9.30	8.17	7.48	3.28	0.113	0.083
	(S*KSi)		6.58	6.06	75.60	38.11	4.11	1.95	8.40	6.49	6.83	3.35	0.137	0.065
	(A*KSi)		6.58	6.06	75.60	38.11	4.11	1.95	8.40	6.49	6.83	3.35	0.137	0.065
	(S*A*KSi)		11.40	10.49	130.95	66.00	7.13	3.38	14.55	11.24	11.84	5.80	0.238	0.113

S-I: 1st season, S-II: 2nd season. Values with the same letter within a column for each treatment are not significantly (*p* > 0.05) different, according to Duncan’s multiple range test.

**Table 5 plants-11-00497-t005:** Effect of foliar spraying with aloe extract (Ae), potassium silicate (KSi) under different salinity levels and their interaction on photosynthetic efficiency, relative water content, and membrane stability index of roselle plants.

Salinity Levels	Ae %	KSi g L^−1^	Fv/Fm		PI		SPAD		RWC %		MSI %	
			S-I	S-II	S-I	S-II	S-I	S-II	S-I	S-II	S-I	S-II
S1 (ECe < 4 dS/m)	0	0	0.73 ^c–g^	0.73 ^c–h^	4.27 ^a–d^	4.42 ^ab^	36.90 ^h^	39.63 ^e–h^	73.38 ^de^	72.87 ^cd^	63.60 ^e–h^	62.14 ^de^
		30	0.77 ^a–d^	0.75 ^a–g^	4.39 ^a–c^	4.62 ^ab^	48.33 ^a–d^	50.03 ^ab^	78.74 ^a–e^	79.28 ^a–d^	68.46 ^a–h^	68.22 ^a–e^
		60	0.78 ^abc^	0.76 ^a–f^	4.73 ^ab^	4.61 ^ab^	41.07 ^c–h^	41.00 ^d–h^	81.79 ^a–d^	81.89 ^ab^	72.98 ^a–d^	70.16 ^a–d^
	0.5	0	0.72 ^efg^	0.74 ^a–h^	2.76 ^e–k^	2.46 ^d–i^	37.93 ^gh^	39.67 ^e–h^	76.60 ^a–e^	76.32 ^a–d^	65.53 ^b–h^	64.59 ^b–e^
		30	0.79 ^a^	0.78 ^a–d^	3.87 ^b–f^	3.42 ^b–f^	50.00 ^a^	52.10 ^a^	81.71 ^a–d^	80.20 ^abc^	73.97 ^abc^	71.80 ^a–d^
		60	0.76 ^a–e^	0.78 ^a–d^	2.77 ^e–k^	2.48 ^d–i^	41.97 ^a–h^	44.23 ^b–g^	84.62 ^ab^	84.39 ^a^	72.81 ^a–d^	72.74 ^abc^
	1	0	0.77 ^a–d^	0.79 ^ab^	2.68 ^e–k^	2.88 ^c–h^	42.73 ^a–h^	42.73 ^c–h^	77.41 ^a–e^	75.06 ^bcd^	66.29 ^b–h^	66.67 ^a–e^
		30	0.80 ^a^	0.80 ^a^	5.25 ^a^	5.14 ^a^	48.97 ^abc^	48.67 ^abc^	85.74 ^a^	84.47 ^a^	76.63 ^a^	75.32 ^a^
		60	0.79 ^ab^	0.80 ^a^	3.39 ^b–h^	3.65 ^b–e^	46.33 ^a–f^	46.13 ^a–d^	83.10 ^abc^	82.74 ^ab^	73.99 ^ab^	74.34 ^ab^
S2 (ECe 4–8 dS/m)	0	0	0.67 ^h^	0.64 ^i^	1.44 ^k^	1.67 ^hi^	38.70 ^e–h^	37.27 ^h^	71.13 ^e^	70.80 ^d^	60.02 ^h^	62.07 ^de^
		30	0.72 ^d–g^	0.72 ^e–h^	2.38 ^g–k^	2.25 ^f–i^	43.47 ^a–h^	41.97 ^d–h^	75.42 ^b–e^	77.08 ^a–d^	67.64 ^a–h^	65.35 ^a–e^
		60	0.70 ^fgh^	0.68 ^hi^	2.29 ^g–k^	2.23 ^f–i^	39.03 ^e–h^	41.90 ^d–h^	78.79 ^a–e^	77.87 ^a–d^	67.67 ^a–h^	68.14 ^a–e^
	0.5	0	0.70 ^gh^	0.72 ^e–h^	1.64 ^jk^	1.58 ^i^	41.43 ^b–h^	38.87 ^fgh^	72.20 ^e^	73.79 ^bcd^	64.09 ^d–h^	63.39 ^cde^
		30	0.76 ^a–e^	0.76 ^a–f^	2.36 ^g–k^	2.26 ^f–i^	49.67 ^ab^	46.43 ^a–d^	77.30 ^a–e^	79.97 ^abc^	67.19 ^b–h^	68.24 ^a–e^
		60	0.78 ^abc^	0.78 ^a–d^	2.77 ^e–k^	2.40 ^e–i^	43.90 ^a–h^	45.13 ^b–f^	78.51 ^a–e^	80.12 ^abc^	70.73 ^a–e^	69.72 ^a–d^
	1	0	0.69 ^gh^	0.71 ^fgh^	2.06 ^h–k^	1.48 ^i^	37.77 ^gh^	38.60 ^gh^	75.23 ^b–e^	75.80 ^a–d^	64.12 ^d–h^	65.07 ^b–e^
		30	0.76 ^a–e^	0.77 ^a–e^	3.17 ^c–i^	2.99 ^c–g^	43.07 ^a–h^	40.60 ^d–h^	78.23 ^a–e^	78.41 ^a–d^	69.79 ^a–g^	67.35 ^a–e^
		60	0.74 ^b–g^	0.72 ^d–h^	2.11 ^h–k^	2.23 ^f–i^	45.27 ^a–g^	46.53 ^a–d^	79.47 ^a–e^	78.47 ^a–d^	70.36 ^a–f^	69.08 ^a–e^
S3 (ECe 8–12 dS/m)	0	0	0.70 ^gh^	0.69 ^ghi^	1.91 ^ijk^	1.68 ^hi^	37.53 ^gh^	36.53 ^h^	70.80 ^e^	70.47 ^d^	60.78 ^gh^	59.47 ^e^
		30	0.77 ^a–e^	0.76 ^a–f^	2.88 ^e–j^	2.60 ^d–i^	40.40 ^d–h^	39.4 ^e–h^	75.08 ^cde^	75.05 ^bcd^	64.88 ^c–h^	65.73 ^a–e^
		60	0.75 ^a–f^	0.76 ^a–f^	2.57 ^f–k^	2.38 ^e–i^	44.63 ^a–h^	42.63 ^c–h^	76.79 ^a–e^	76.87 ^a–d^	67.15 ^b–h^	66.54 ^a–e^
	0.5	0	0.74 ^b–g^	0.73 ^b–h^	1.86 ^i–k^	1.50 ^i^	38.50 ^fgh^	40.93 ^d–h^	71.54 ^e^	71.79 ^cd^	61.33 ^fgh^	61.72 ^de^
		30	0.79 ^ab^	0.78 ^a–d^	3.96 ^a–e^	3.99 ^abc^	47.00 ^a–e^	45.40 ^b–e^	76.97 ^a–e^	76.64 ^a–d^	68.03 ^a–h^	67.41 ^a–e^
		60	0.78 ^abc^	0.77 ^a–e^	3.57 ^b–g^	3.47 ^b–f^	42.80 ^a–h^	42.47 ^c–h^	78.37 ^a–e^	77.59 ^a–d^	70.64 ^a–e^	69.92 ^a–d^
	1	0	0.74 ^b–g^	0.75 ^a–f^	1.79 ^jk^	1.76 ^ghi^	39.20 ^e–h^	38.77 ^gh^	72.12 ^e^	72.47 ^cd^	62.63 ^e–h^	63.41 ^cde^
		30	0.78 ^abc^	0.77 ^a–f^	3.92 ^a–f^	3.71 ^bcd^	45.30 ^a–g^	44.63 ^b–g^	78.03 ^a–e^	78.41 ^a–d^	66.69 ^b–h^	67.67 ^a–e^
		60	0.80 ^a^	0.79 ^abc^	2.91 ^d–j^	2.70 ^d–i^	49.43 ^ab^	46.83 ^a–d^	79.14 ^a–e^	78.47 ^a–d^	67.85 ^a–h^	68.78 ^a–e^
LSD 5%	(S)		0.03	0.03	1.04	0.31	5.31	5.77	1.92	4.08	1.62	8.17
	(A)		0.02	0.02	0.45	0.35	3.26	4.16	2.83	4.57	2.94	3.82
	(KSi)		0.02	0.02	0.46	0.43	2.77	2.12	3.15	2.99	3.03	3.38
	(S*A)		0.03	0.03	0.78	0.6	5.68	7.21	4.90	7.92	5.08	6.62
	(S*KSi)		0.03	0.04	0.80	0.74	4.80	3.67	5.46	5.18	5.24	5.86
	(A*KSi)		0.03	0.04	0.80	0.74	4.80	3.67	5.45	5.19	5.25	5.87
	(S*A*KSi)		0.05	0.06	1.38	1.28	8.32	6.36	9.45	8.98	9.09	10.14

Relative water content (RWC), membrane stability index (MSI), single-photon avalanche diode (SPAD), performance index (PI) of photosynthesis, maximum light-adaptive fluorescence (Fm), variable fluorescence (Fv). S-I: 1st season, S-II: 2nd season. Values with the same letter within a column for each treatment are not significantly (*p* > 0.05) different, according to Duncan’s multiple range test.

**Table 6 plants-11-00497-t006:** Effect of foliar spraying with Aloe extract (Ae) and potassium silicate (KSi) and their interaction on total soluble sugars, total free amino acids, and nutrient contents in roselle plants under different salinity levels.

Salinity Levels	Ae %	KSi g L^−1^	TSS (mg g^−1^ DW)		TFAA (mg g^−1^ DW)		Proline (µ mole g^−1^)		K^+^ (mg g^−1^ DW)		Na^+^ (mg g^−1^ DW)		SiO_2_ (mg g^−1^ DW)		Cl^−^ (mg g^−1^ DW)	
			S-I	S-II	S-I	S-II	S-I	S-II	S-I	S-II	S-I	S-II	S-I	S-II	S-I	S-II
S1 (ECe < 4 dS/m)	0	0	343.30 ^cd^	343.30 ^cde^	73.90 ^klm^	75.30 ^gh^	0.23 ^jkl^	0.22 ^g–k^	6.63 ^lm^	6.79 ^i–m^	0.49 ^gh^	0.48 ^i^	15.80 ^g^	14.70 ^k^	2.88 ^f^	3.76 ^j^
		30	417.73 ^cd^	350.00 ^cde^	89.10 ^g^	86.30 ^fg^	0.26 ^hij^	0.25 ^f–i^	6.87 ^ij^	6.85 ^h–l^	0.44 ^ij^	0.44 ^j^	23.00 ^bcd^	22.70 ^b^	2.88 ^f^	3.09 ^i^
		60	435.10 ^bc^	370.00 ^cd^	132.00 ^b^	138.10 ^b^	0.33 ^e^	0.34 ^de^	7.57 ^b^	7.13 ^c–g^	0.41 ^jk^	0.41 ^jk^	23.20 ^bcd^	22.80 ^b^	2.88 ^f^	3.09 ^i^
	0.5	0	396.90 ^de^	343.30 ^c–f^	67.40 ^n^	70.80 ^fgh^	0.44 ^c^	0.48 ^c^	6.70 ^l^	6.63 ^klm^	0.41 ^jk^	0.41 ^jkk^	20.30 ^e^	19.30 ^hi^	2.88 ^f^	3.09 ^i^
		30	495.70 ^a^	460.00 ^b^	110.80 ^d^	127.20 ^cd^	0.21 ^lm^	0.20 ^j–m^	7.37 ^d^	7.25 ^b–e^	0.41 ^jk^	0.41 ^k^	22.10 ^cde^	21.20 ^de^	2.85 ^f^	3.09 ^i^
		60	516.70 ^a^	553.30 ^a^	152.70 ^a^	157.70 ^a^	0.29 ^fg^	0.31 ^e^	8.13 ^a^	8.05 ^a^	0.37 ^l^	0.36 ^l^	23.90 ^abc^	22.40 ^bc^	2.82 ^f^	2.73 ^j^
	1	0	374.40 ^e^	348.40 ^f^	92.40 ^g^	96.70 ^f^	0.24 ^ijk^	0.23 ^f–j^	6.94 ^hi^	6.60 ^lm^	0.41 ^jk^	0.41 ^jk^	21.30 ^de^	20.10 ^fgh^	2.88 ^f^	3.09 ^i^
		30	436.70 ^b^	377.00 ^c^	104.10 ^ef^	106.30 ^e^	0.70 ^b^	0.71 ^b^	7.00 ^gh^	6.87 ^g–k^	0.39 ^kl^	0.38 ^kl^	24.20 ^ab^	23.30 ^b^	2.49 ^g^	2.54 ^k^
		60	447.20 ^b^	436.70 ^b^	116.80 ^c^	117.20 ^de^	0.77 ^a^	0.75 ^a^	7.51 ^bc^	7.51 ^b^	0.36 ^l^	0.36 ^l^	25.40 ^a^	25.30 ^a^	2.49 ^g^	2.54 ^k^
S2 (ECe 4–8 dS/m)	0	0	303.33 ^h–l^	306.00 ^e–i^	68.50 ^mn^	67.50 ^h^	0.18 ^n^	0.18 ^mn^	6.10 ^n^	6.01 ^n^	0.55 ^de^	0.55 ^def^	11.90 ^h^	11.30 ^l^	3.57 ^d^	3.70 ^de^
		30	317.10 ^g–j^	313.30 ^e–h^	90.20 ^g^	82.90 ^fgh^	0.25 ^hij^	0.25 ^fgh^	6.59 ^m^	6.57 ^m^	0.54 ^ef^	0.54 ^efg^	21.00 ^e^	20.00 ^fgh^	3.55 ^d^	3.61 ^ef^
		60	318.90 ^ghi^	317.20 ^g–j^	97.30 ^h–k^	95.50 ^f^	0.24 ^ijk^	0.26 ^fg^	7.21 ^e^	7.06 ^d–h^	0.54 ^ef^	0.53 ^fgh^	20.30 ^e^	20.30 ^efg^	3.55 ^d^	3.61 ^ef^
	0.5	0	306.80 ^h–k^	306.70 ^e–i^	80.00 ^h^	83.60 ^fg^	0.21 ^lm^	0.22 ^h–l^	6.79 ^k^	6.74 ^j–m^	0.55 ^def^	0.54 ^efg^	18.50 ^e^	17.50 ^j^	3.55 ^d^	3.61 ^ef^
		30	318.90 ^ghi^	319.00 ^d–g^	98.90 ^f^	96.10 ^f^	0.25 ^h–k^	0.23 ^f–j^	7.10 ^f^	7.01 ^e–i^	0.51 ^fg^	0.51 ^hi^	21.80 ^de^	20.90 ^def^	3.36 ^g^	3.53 ^f^
		60	324.10 ^gh^	320.70 ^d–g^	104.30 ^e^	112.60 ^de^	0.26 ^ghi^	0.26 ^fg^	7.30 ^d^	7.16 ^c–f^	0.49 ^gh^	0.50 ^i^	22.10 ^cde^	21.20 ^de^	3.30 ^gh^	3.30 ^gh^
	1	0	315.50 ^g–j^	306.80 ^h–k^	74.40 ^jkl^	75.70 ^gh^	0.22 ^kl^	0.21 ^i–m^	6.84 ^jk^	6.70 ^j–m^	0.51 ^fg^	0.51 ^ghi^	21.70 ^de^	19.80 ^gh^	3.54 ^d^	3.54 ^d^
		30	329.70 ^c–g^	323.30 ^d–g^	109.80 ^d^	106.70 ^e^	0.26 ^hij^	0.25 ^fgh^	7.04 ^fg^	6.90 ^f–j^	0.49 ^gh^	0.48 ^i^	22.10 ^cde^	21.20 ^de^	3.21 ^e^	3.22 ^h^
		60	334.50 ^fg^	327.70 ^d–g^	132.60 ^b^	144.00 ^ab^	0.27 ^f–i^	0.26 ^f^	7.47 ^c^	7.33 ^bcd^	0.45 ^hi^	0.48 ^i^	22.50 ^bcd^	21.40 ^d^	3.19 ^e^	3.21 ^e^
S3 (ECe 8–12 dS/m)	0	0	201.10 ^q^	226.70 ^j^	67.00 ^n^	67.00 ^h^	0.09 ^o^	0.10 ^p^	5.46 ^q^	5.42 ^p^	0.68 ^a^	0.67 ^a^	10.90 ^h^	10.40 ^l^	5.96 ^a^	5.73 ^a^
		30	251.30 ^op^	266.00 ^hij^	81.00 ^h^	85.00 ^fg^	0.19 ^mn^	0.19 ^k–n^	5.80 ^1p^	5.72 ^o^	0.58 ^cd^	0.57 ^cd^	17.50 ^fg^	16.60 ^j^	4.23 ^b^	4.47 ^b^
		60	261.70 ^nop^	266.70 ^hij^	75.30 ^ijkl^	76.30 ^gh^	0.19 ^mn^	0.18 ^lmn^	6.12 ^n^	6.06 ^n^	0.58 ^cde^	0.56 ^cde^	22.40 ^bcd^	21.60 ^cd^	3.93 ^c^	4.14 ^c^
	0.5	0	241.10 ^p^	261.00 ^ij^	79.90 ^hij^	81.30 ^fgh^	0.17 ^n^	0.16 ^no^	5.89 ^o^	5.78 ^no^	0.63 ^b^	0.60 ^b^	18.50 ^ef^	17.50 ^j^	4.29 ^b^	4.39 ^b^
		30	268.70 ^mno^	283.30 ^ghi^	90.70 ^g^	95.60 ^f^	0.28 ^fgh^	0.23 ^f–k^	6.59 ^m^	6.57 ^m^	0.58 ^c^	0.58 ^bc^	18.50 ^ef^	17.50 ^j^	3.93 ^c^	4.08 ^c^
		60	280.80 ^lmn^	290.00 ^f–i^	106.00 ^de^	105.70 ^e^	0.45 ^c^	0.45 ^c^	7.06 ^fg^	7.02 ^e–i^	0.55 ^cde^	0.58 ^bc^	21.30 ^de^	20.20 ^fg^	3.87 ^c^	3.75 ^d^
	1	0	246.70 ^op^	263.30 ^hij^	73.90 ^lm^	75.20 ^gh^	0.09 ^o^	0.11 ^p^	6.94 ^hi^	7.09 ^d–h^	0.59 ^bc^	0.57 ^cd^	20.40 ^e^	19.30 ^hi^	4.27 ^b^	4.08 ^c^
		30	289.50 ^klm^	298.10 ^i–l^	80.40 ^hi^	84.90 ^fg^	0.29 ^f^	0.13 ^op^	7.52 ^bc^	7.38 ^bc^	0.58 ^cd^	0.57 ^cd^	21.20 ^e^	18.90 ^i^	3.57 ^d^	3.70 ^de^
		60	294.70 ^jkl^	300.00 ^e–i^	82.00 ^h^	86.40 ^fg^	0.40 ^d^	0.38 ^d^	7.10 ^f^	7.10 ^d–h^	0.58 ^cde^	0.55 ^def^	21.30 ^de^	20.10 ^fgh^	3.57 ^d^	3.70 ^de^
LSD 5%	(S)		3.21	12.18	2.11	11.13	0.02	0.10	0.59	0.26	0.03	0.022	0.04	0.07	0.05	0.032
	(A)		8.78	13.07	2.01	9.02	0.08	0.01	0.43	0.26	0.02	0.011	0.04	0.14	0.03	0.029
	(KSi)		7.57	16.68	1.55	5.58	0.08	0.01	0.64	0.33	0.01	0.080	0.02	0.08	0.03	0.031
	(S*A)		12.81	22.63	3.48	15.62	0.01	0.02	0.76	0.45	0.03	0.019	0.06	0.24	0.06	0.052
	(S*KSi)		11.15	28.89	2.69	9.67	0.02	0.02	1.11	0.57	0.02	0.014	0.04	0.13	0.06	0.053
	(A*KSi)		13.84	28.89	2.68	9.67	0.02	0.02	1.11	0.57	0.02	0.014	0.04	0.13	0.06	0.053
	S*A*KSi		22.72	50.04	4.65	16.75	0.03	0.04	1.92	0.99	0.04	0.024	0.06	0.23	0.10	0.092

S-I: 1st season, S-II: 2nd season. Total soluble sugars (TSS), total free amino acids (TFAA), K^+^, potassium, Na^+^, sodium, SiO_2_, silicon dioxide “silica”, Cl^−^, chloride, dry weight (DW). Values with the same letter within a column for each treatment are not significantly (*p* > 0.05) different, according to Duncan’s multiple range test.

**Table 7 plants-11-00497-t007:** Effect of foliar spraying with aloe extract (Ae) and potassium silicate (KSi), and their interaction on physicochemical characteristics of roselle calyces under different salinity levels.

Salinity Levels	Ae %	KSi g L^−1^	Anthocyanin (mg g^−1^ DW)		Total Soluble Sugars (%)		Acidity Citric Acid %		pH	
			S-I	S-II	S-I	S-II	S-I	S-II	S-I	S-II
S1 (ECe < 4 dS/m)	0	0	94.31 ^g^	88.30 ^n^	0.36 ^cd^	0.35 ^d^	0.62 ^h^	0.63 ^ghi^	2.89 ^de^	2.91 ^def^
		30	95.40 ^fg^	90.90 ^m^	0.36 ^cd^	0.36 ^c^	0.63 ^gh^	0.63 ^ghi^	2.98 ^bc^	2.98 ^b^
		60	104.90 ^de^	97.40 ^k^	0.41 ^a^	0.40 ^b^	0.68 ^e^	0.67 ^e^	2.94 ^cd^	2.95 ^cde^
	0.5	0	121.20 ^c^	119.90 ^d^	0.36 ^cd^	0.36 ^c^	0.72 ^d^	0.73 ^d^	2.88 ^e^	2.90 ^efg^
		30	121.90 ^c^	121.30 ^d^	0.41 ^a^	0.40 ^b^	0.72 ^d^	0.73 ^d^	2.82 ^f^	2.84 ^hi^
		60	152.40 ^a^	147.90 ^a^	0.41 ^a^	0.40 ^b^	0.74 ^d^	0.74 ^d^	2.72 ^gh^	2.76 ^klm^
	1	0	107.10 ^de^	102.30 ^i^	0.35 ^d^	0.35 ^d^	0.85 ^b^	0.86 ^bc^	2.68 ^hi^	2.75 ^lm^
		30	108.30 ^d^	104.60 ^h^	0.41 ^a^	0.40 ^b^	0.86 ^ab^	0.87 ^ab^	2.66 ^i^	2.72 ^m^
		60	122.40 ^c^	123.40 ^c^	0.42 ^a^	0.43 ^a^	0.88 ^a^	0.88 ^a^	2.50 ^j^	2.59 ^n^
S2 (ECe 4–8 dS/m)	0	0	87.50 ^h^	86.30 ^o^	0.36 ^cd^	0.35 ^d^	0.61 ^h^	0.63 ^ghi^	2.85 ^ef^	2.85 ^ghi^
		30	94.70 ^g^	88.70 ^n^	0.36 ^cd^	0.36 ^c^	0.64 ^fg^	0.63 ^ghi^	2.84 ^ef^	2.86 ^fghi^
		60	104.50 ^de^	95.30 ^l^	0.41 ^a^	0.40 ^b^	0.64 ^fg^	0.64 ^ghi^	2.73 ^g^	2.78 ^j^
	0.5	0	106.70 ^de^	100.30 ^j^	0.36 ^cd^	0.36 ^c^	0.61 ^h^	0.62 ^hi^	2.75 ^g^	2.89 ^fgh^
		30	106.80 ^de^	100.90 ^j^	0.41 ^a^	0.40 ^b^	0.65 ^f^	0.65 ^fg^	2.67 ^hi^	2.71 ^m^
		60	121.30 ^c^	119.50 ^e^	0.41 ^a^	0.40 ^b^	0.67 ^e^	0.67 ^e^	2.52 ^j^	2.59 ^n^
	1	0	120.30 ^c^	118.00 ^ef^	0.36 ^cd^	0.36 ^c^	0.83 ^c^	0.84^c^	2.76 ^g^	2.80 ^i–l^
		30	120.70 ^c^	118.10 ^ef^	0.41 ^a^	0.40 ^b^	0.83 ^c^	0.84 ^c^	3.01 ^ab^	2.99 ^bc^
		60	151.20 ^a^	145.60 ^b^	0.41 ^a^	0.41 ^b^	0.87 ^ab^	0.85 ^bc^	2.87 ^ef^	2.89 ^fgh^
S3 (ECe 8–12 dS/m)	0	0	93.00 ^g^	85.30 ^o^	0.29 ^g^	0.28 ^h^	0.56 ^j^	0.56 ^k^	3.06 ^a^	3.05 ^a^
		30	93.90 ^g^	87.60 ^no^	0.29 ^g^	0.30 ^g^	0.59 ^i^	0.61 ^ij^	2.98 ^bc^	2.97 ^bc^
		60	103.70 ^e^	94.40 ^l^	0.32 ^f^	0.31 ^fg^	0.65 ^f^	0.66 ^ef^	2.97 ^bc^	2.97 ^bc^
	0.5	0	120.10 ^c^	117.40 ^fg^	0.33 ^e^	0.32 ^f^	0.64 ^fg^	0.64 ^ghi^	3.03 ^ab^	3.02 ^ab^
		30	120.30 ^c^	118.10 ^ef^	0.34 ^e^	0.34 ^e^	0.65 ^f^	0.65 ^fg^	2.68 ^h^	2.73 ^m^
		60	144.70 ^b^	145.40 ^b^	0.38 ^b^	0.37 ^c^	0.65 ^f^	0.63 ^ghi^	2.54 ^j^	2.63 ^n^
	1	0	99.20 ^f^	99.60 ^j^	0.36 ^cd^	0.36 ^c^	0.72 ^d^	0.73 ^d^	2.86 ^ef^	2.89 ^fgh^
		30	105.90 ^de^	99.40 ^j^	0.36 ^cd^	0.36 ^c^	0.73 ^d^	0.74 ^d^	2.85 ^ef^	2.87 ^fgh^
		60	120.80 ^c^	119.50 ^e^	0.37 ^c^	0.36 ^c^	0.73 ^d^	0.73 ^d^	2.75 ^g^	2.81 ^ijk^
LSD 5%	(S)		1.34	3.42	0.03	0.04	0.09	0.08	0.01	0.05
	(A)		1.87	0.82	0.06	0.08	0.04	0.03	0.06	0.06
	(KSi)		1.03	0.50	0.06	0.06	0.02	0.04	0.02	0.02
	(S*A)		3.24	1.42	0.01	0.01	0.07	0.05	0.10	0.11
	(S*KSi)		1.78	0.86	0.01	0.01	0.05	0.06	0.03	0.03
	(A*KSi)		1.78	0.86	0.01	0.01	0.05	0.07	0.03	0.03
	S*A*KSi		3.08	1.49	0.02	0.02	0.08	0.01	0.05	0.05

S-I: 1st season, S-II: 2nd season. Values with the same letter within a column for each treatment are not significantly (*p* > 0.05) different, according to Duncan’s multiple range test.

## Data Availability

Data is contained within the article.

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
