# Peer review of "Improvement of Selected Morphological, Physiological, and Biochemical Parameters of Roselle (Hibiscus sabdariffa L.) Grown under Different Salinity Levels Using Potassium Silicate and Aloe saponaria Extract"

_plants, 2022, doi:10.3390/plants11040497_

Round 1

Reviewer 1 Report

I would like to make the corrections that I mentioned in the MS.

Author Response

Reviewer 1 comments

Page 1

long title, I would recommend to modify this

Response: it was adjusted accordingly

remove from the Abstract and move it into Methods

Response: Done accordingly

Please mention the full name then Abbreviation

Response: Done accordingly

Page 4

I would suggest to move this table in the results section.

Response: Done accordingly

Page 5

transfer it into results section

Response: Done accordingly

Page 8, 14, 15

Delete

Response: Done accordingly

Page 9

Please mention the type of test that you are using

Response: Done accordingly

already mentioned in the method section, so remove please.

Response: Done accordingly

Page 10

Please mention the type of test

Response: Done accordingly

Delete

Response: Done accordingly

Page 13

once is enough

Response: Done accordingly

once mention is enough in the end of paragraph

Response: Done accordingly

Delete

Response: Done accordingly

Page 14, 15

Delete

Response: Done accordingly

Page 16

delete from this section and move it into introduction

Response: Done accordingly

Page 17

Negative

Response: Done accordingly

Page 18

need more clarification

Response: Done accordingly

page 19

Please add some interested results in the end of this section.

Response: Done accordingly

Reviewer 2 Report

The experiment is interesting and its results may have practical applications.
In the introduction, I recommend adding the objectives of the experiment.
The presentation of the results in tables is not clear. Also their description in the text. I recommend a change. The discussion is very extensive, but the actual results are relatively little discussed.
How did the authors arrive at the recommended concentrations given in the conclusion?
I recommend a careful review of the text. See my comments on attached pdf file.

Author Response

Reviewer 2 comments

Thanks for the reviewer, the comments significantly enhanced the manuscript. We have replied all comments accordingly

Page 1

What does cml-1 mean? I recommend adjusting the unit.

Response: it means cm/L

dS m-1, Edit everywhere in the text. I recommend checking the units according to the editorial requirements

Response: Done accordingly

Fv/Fm

Response: Done accordingly

Page 2

is not correct as follows .... of 3 cml-1 of KSi plus 1% of Ae with S1?

Response: it was corrected accordingly

I recommend starting the text as a new paragraph.

Response: Done accordingly

Page 4

I recommend starting the text as a new paragraph.

Response: Done accordingly

I think we need to add to the objectives of the experiment

Response: Done accordingly

correct 34.8 °C

Response: Done accordingly

Page 5

correctly L

Response: Done accordingly

Is there a reference to Table 2 in the text?

Response: Done accordingly

Page 9

What do S-I and S-II mean?

Response: it was observed accordingly

Page 18

What is the relevance of this text to the experiment?

Response: it was detailed in the text accordingly

Reviewer 3 Report

Dear Editor and Authors,

The manuscript received for review raises an interesting topic. However, the presentation of the collected data is so incoherent and chaotic that it is not suitable for publication in this form. The article is not prepared in accordance with the requirements of the journal. The abstract itself contains over 200 words that are required ... The introduction is a uniform text without paragraphs and a marked work goal! The authors present the methodology one by one, which also does not comply with the requirements ... This part is also prepared very briefly, in general ... Results - you don't really know what they are about. A large amount of data, from which little can be read ... The discussion also requires rewriting. References are not tailored to the journal. I do not recomennded this paper to publication in Plant journal.

Author Response

Reviewer 3 comments

Thanks for the reviewer, the comments significantly enhanced the manuscript. We have replied all comments accordingly

Page 1

Improvement of selected morphological, physiological and biochemical parameters of Roselle (Hibiscus sabdariffa L.) Grown under Salinity Modulates Using Potassium Silicate and Aloe Saponaria Extract

Response: Done accordingly

Abstract too long, inconsistent with the manuscript template

Response: it was briefed

Page 2

they are in the title so delete and add anothers

Response: Done accordingly
in my opinion, this part should be shortened. I would remove the sentences about diseases - please mention it in one short sentence

Response: Done accordingly

add latine name here

Response: Done accordingly

from new paragraph PLEASE

Response: Done accordingly

new paragraph

Response: Done accordingly

Page 3

In my opinion it is too long

Response: it was briefed and all structural and type mistakes were corrected accordingly

full latine name (Aloë L.) aloe in italic

Response: Done accordingly

Page 4

Aloe maculata All. (A. saponaria)

Response: Done accordingly

to rewritten...
Response: Done accordingly

I do not see aims of this paper

Response: the aim was added accordingly
here should be presented results please check manuscript template

Response: Done accordingly

Page 5

study, experiment, research ttrial we can use for clinical trials

Response: the verbs were checked and corrected

Page 6

Month we write in big letters!

Response: Done accordingly

Chlorophyll a fluorescence, chlorophyll and performance index measurements

Response: Done accordingly

please add more information about these methods

Response: Done accordingly

only one parameter, why? I suggest add more parameters!

Response: Done accordingly

How did you do these measurements? Please describe them.

Response: it was detailed accordingly

add more information about these methodology

Response: it was detailed accordingly

Page 7

None of the tables are prepared correctly. The presentation of the results is so chaotic that it is difficult to find out anything. Each of the tables needs to be refined. It is not acceptable in this form.

Response: Thanks for this valuable comment, all tables were reformulated and the results were detailed accordingly

Page 19

References are not prepared according to manuscript template
Response: Done accordingly

Round 2

Reviewer 2 Report

The authors have carefully processed all comments. The quality of the manuscript has increased significantly. I have no further comments.

Author Response

Thanks to the reviewer for the positive comments which enhanced the manuscript

Reviewer 3 Report

Dear Editor and Authors,

The authors have partially complied with my comments. The article still needs to be refined. I do not consent to the publication in this form.

Author Response

Reviewer 3 comments

The authors have partially complied with my comments. The article still needs to be refined. I do not consent to the publication in this form.

Thanks to the reviewer, the comments significantly enhanced the manuscript. We have revised again the comments

Page 1

Improvement of selected morphological, physiological and biochemical parameters of Roselle (Hibiscus sabdariffa L.) Grown under Salinity Modulates Using Potassium Silicate and Aloe Saponaria Extract

Response: Done accordingly

Abstract too long, inconsistent with the manuscript template

Response: it was briefed

Page 2

They are in the title so delete and add another’s

Response: Done accordingly
in my opinion, this part should be shortened. I would remove the sentences about diseases - please mention it in one short sentence

Response: Done accordingly

add latine name here

Response: Done accordingly

from new paragraph PLEASE

Response: Done accordingly

new paragraph

Response: Done accordingly

Page 3

In my opinion it is too long

Response: it was briefed and all structural and type mistakes were corrected accordingly

full latine name (Aloë L.) aloe in italic

Response: Done accordingly

Page 4

Regarding The table of Soil analysis

Response: Thanks for this comment, usually this table occurred in materials and methods

Aloe maculata All. (A. saponaria)

Response: Done accordingly

to rewritten...
Response: Done accordingly

I do not see aims of this paper

Response: the aim was added accordingly
here should be presented results please check manuscript template

Response: Done accordingly

Page 5

study, experiment, research ttrial we can use for clinical trials

Response: the verbs were checked and corrected

Page 6

Month we write in big letters!

Response: Done accordingly

Chlorophyll a fluorescence, chlorophyll and performance index measurements

Response: Done accordingly

please add more information about these methods

Response: Done accordingly

only one parameter, why? I suggest add more parameters!

Response: Done accordingly

How did you do these measurements? Please describe them.

Response: it was detailed accordingly

add more information about these methodology

Response: it was detailed accordingly

Page 7

None of the tables are prepared correctly. The presentation of the results is so chaotic that it is difficult to find out anything. Each of the tables needs to be refined. It is not acceptable in this form.

Response: Thanks for this valuable comment, all tables were reformulated and the results were detailed accordingly

Page 19

References are not prepared according to manuscript template
Response: Done accordingly